# Auxiliary Classifiers Improve Stability and Efficiency in Continual Learning

## Abstract

Continual learning is crucial for applications in dynamic environments, where machine learning models must adapt to changing data distributions while retaining knowledge of previous tasks. Despite significant advancements, catastrophic forgetting — where performance on earlier tasks degrades as new information is learned — remains a key challenge. In this work, we investigate the stability of intermediate neural network layers during continual learning and explore how auxiliary classifiers (ACs) can leverage this stability to improve performance. We show that early network layers remain more stable during learning, particularly for older tasks, and that ACs applied to these layers can outperform standard classifiers on past tasks. By integrating ACs into several continual learning algorithms, we demonstrate consistent and significant performance improvements on standard benchmarks. Additionally, we explore dynamic inference, showing that AC-augmented continual learning methods can reduce computational costs by up to 60% while maintaining or exceeding the accuracy of standard methods. Our findings suggest that ACs offer a promising avenue for enhancing continual learning models, providing both improved performance and the ability to adapt the network computation in environments where such flexibility might be required.

## 1 Introduction

The field of continual learning provides theories and algorithms for learning from non-i.i.d. data streams (De Lange et al., 2021). The most commonly studied scenario involves data arriving in sequences of tasks, where the learner cannot access previously seen tasks when learning new ones. Continual learning scenarios may involve tasks with different data distributions (domain-incremental learning) or new classes (class-incremental learning) and also vary based on whether task identity is available during classification (task-incremental learning) (Van de Ven & Tolias, 2019). The primary challenge in continual learning, *catastrophic forgetting*, refers to a significant drop in performance on past tasks throughout the learning (McCloskey & Cohen, 1989; Kirkpatrick et al., 2017). Various strategies have been proposed to address this challenge, including parameter isolation (Rusu et al., 2016; Serra et al., 2018; Mallya & Lazebnik, 2018), weight and data regularization (Aljundi et al., 2018; Kirkpatrick et al., 2017; Li & Hoiem, 2017), and rehearsal methods (Rebuffi et al., 2017; Chaudhry et al., 2018). Despite these efforts, continual learning remains an open problem, especially in the widely applicable class-incremental setting that is the focus of our work.

Several works have observed that continual learning mainly results in changes in the later layers of the network (Liu et al., 2020a; Ramasesh et al., 2020; Zhao et al., 2023) and that deep networks trained for image classification split into parts that build their representations differently (Masarczyk et al., 2023). However, these works do not exploit these observations to improve the performance. In this paper, we analyze whether the higher stability of intermediate layers can be leveraged to improve accuracy on previous tasks. First, we examine the stability of representations at different network levels during continual learning, confirming that early layers change less during continual learning, especially for the old data. Next, we evaluate the performance of auxiliary classifiers (ACs) learned on top of such representations through linear probing and show that they perform comparable or even better than the final network classifier on older tasks. We also examine the diversity of the prediction across the added classifiers and demonstrate that they learn to classify different subsets of data, with some samples being correctly predicted at only a single intermediate layer. Finally, we compare the performance of multi-classifier networks with ACs trained jointly and separately

Figure 1: CKA of the first task representations across different ResNet32 layers (L1.B3-L3.B5) through continual learning on CIFAR100 split into 10 tasks. Representations at the early layers are more similar across the continual learning, hinting at the potential for more stability that could be leveraged to improve the performance.

with the rest of the model modules and show that joint training improves the performance of early classifiers with almost no negative effect on the later ones.

Motivated by the findings from our analysis, we advocate for the use of ACs in continual learning. We enhance various standard continual learning methods (LwF (Li & Hoiem, 2017), EWC (Kirkpatrick et al., 2017), ER (Riemer et al., 2018), BiC (Wu et al., 2019), SSIL (Ahn et al., 2021), ANCL (Kim et al., 2023), LODE (Liang & Li, 2024)) with the ACs and show that by combining the predictions from multiple classifiers we can robustly outperform a standard, single-classifier network on standard benchmarks such as CIFAR100 and ImageNet100 on equally-sized tasks and in the warm-start scenario (Magistri et al.; Goswami et al., 2024). Inspired by early-exit literature (Panda et al., 2016; Teerapittayanon et al., 2016; Kaya et al., 2019), we also experiment with dynamic inference in AC-based networks that enables the user to adapt the average network computation to the available resources without any additional training. We show that AC networks used with such inference can maintain the performance of the single-classifier baseline while using only 40-70% of the original network computation. We perform a thorough ablation study of architectural modifications of AC networks and show that our approach robustly improves performance across all tested cases and does not require any meticulous hyperparameter optimization. Our work demonstrates that continual methods enhanced with ACs exhibit better stability in continual learning, achieve higher accuracy, and can be an alternative to standard models in scenarios that require faster inference or the ability to control the compute in the network. The main contributions of our work are:

- We perform a thorough analysis of intermediate representations in continual learning and show that they enable learning diverse classifiers that perform well on different subsets of data. We show that early representations are more stable and the classifiers learned on top of such representations are less prone to forgetting the older tasks.

- We leverage the diversity and robustness of intermediate representations by enhancing the networks with auxiliary classifiers (ACs). We integrate ACs into several continual learning methods and demonstrate that AC-enhanced methods consistently outperform standard single-classifier approaches, achieving an average 10% relative improvement.

- We show that ACs can help reduce the average computational cost during network inference through dynamic prediction. AC-enhanced methods can achieve similar accuracy to single-classifier models while using only 40-70% of the computational resources.

## 2 INTERMEDIATE LAYER REPRESENTATIONS IN CONTINUAL LEARNING

In this section, we analyze the stability of intermediate representations in continual learning and the use of *auxiliary classifiers* (ACs) - additional classifiers trained on to the intermediate representations of the network - as a means to leverage this stability. We consider a supervised continual learning scenario, where a learner (neural network) is trained over $T$ classification tasks and its goal is to learn to classify the new classes while avoiding catastrophic forgetting of the previously learned ones. We focus on the more challenging class-incremental learning setting (De Lange et al., 2021; Masana et al., 2022), where the learner needs to distinguish between all the classes encountered so

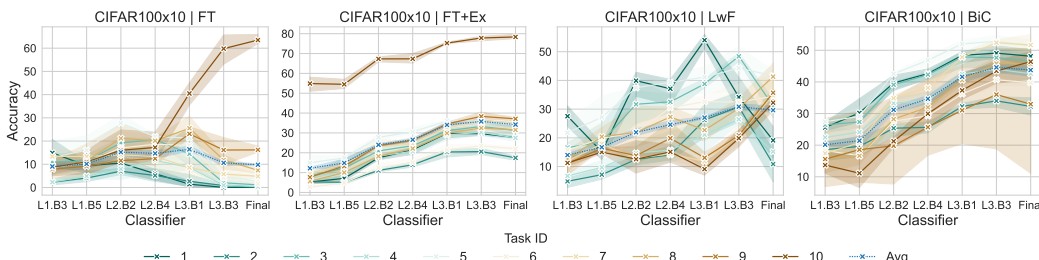

Figure 2: Per-task final accuracy of the auxiliary classifiers trained with linear probing on top of several network layers and final network classifier on CIFAR100 split into 10 tasks. For most tasks, some of the auxiliary classifiers outperform the final classifiers, as higher stability of intermediate representations across the training leads to reduced forgetting.

far without having access to a task identity. At each task $t$, the model can only access the dataset $\mathcal{D}_t = \{\mathcal{X}_t, \mathcal{Y}_t\}$, which is composed of a set of input images $\mathcal{X}_t$ and corresponding labels $\mathcal{Y}_t$. We analyze an offline learning scenario, where the learner can pass through the data samples from the current task multiple times.

For our initial analysis, we conduct experiments on CIFAR100 (Krizhevsky, 2009). We present most of the results on the 10-task split and include corresponding results on the 5-task split in Appendix B, as they are very similar. We consider naive *finetuning (FT)* scenario without any additional continual learning technique and standard continual learning methods such as finetuning with exemplars (FT+Ex), exemplar-free LwF (Li & Hoiem, 2017) and BiC (Wu et al., 2019). We believe this set of methods to be a good overview across the continual learning method landscape, as they involve either replay, regularization or both. In the setting with exemplars, we utilize a memory buffer to store part of the training data and for each task $t > 1$ we train the model on the original dataset $\mathcal{D}_t$ extended with the exemplar samples from the memory. We keep the size of the memory buffer fixed and update it after we finish training on each task. Refer to the Section 4 and Appendix I for more details on our experimental setup.

## 2.1 STABILITY OF INTERMEDIATE REPRESENTATIONS

We begin our investigation by analyzing the stability of the representations at the different layers of ResNet32 (He et al., 2016) over the course of continual learning on CIFAR100. We investigate the stability through similarity between the original representations of the first task data learned after the first task and the representations of this data after learning each subsequent task $t$. We select a subset of 6 intermediate layers (L1.B3-L3.B3) uniformly spread by the compute similarly to Kaya et al. (2019) alongside the final feature layer L3.B5 that precedes the classifier and present their representational similarity measured with CKA (Kornblith et al., 2019) in Figure 1.

Consistently with previous research, we observe that the early layer representations change less and exhibit more stability through the learning phase. In contradiction, the final layer representations usually change the most during the training, and the phenomenon gets stronger if we train on more tasks. While continual learning methods such as FT+Ex, LwF, or BiC are more stable than naive FT, the trend of CKA increasing for early layer representations persists. The higher stability of early representations indicates the potential for their use in continual learning, as we can expect them to be less prone to forgetting.

## 2.2 MEASURING INTERMEDIATE REPRESENTATION QUALITY WITH LINEAR PROBING

Representational similarity across tasks might not directly translate to strong continual learning performance. To assess whether intermediate representations are suitable for class-incremental learning, we employ linear probing (Davari et al., 2022), a well-known technique used to measure the quality of representations through a downstream performance of auxiliary classifiers (ACs) continually trained on intermediate representations (without gradient propagation from the classifiers to the original network). For ACs, we use a simple pooling layer to reduce the feature dimensionality and

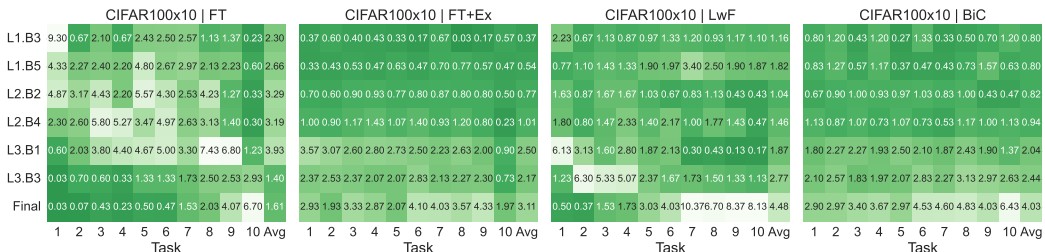

Figure 3: Unique accuracy (a subset of samples that a single given classifier classifies correctly) of auxiliary classifiers and final network classifier for different task data on CIFAR100 split into 10 tasks. Classifiers built on the intermediate representations enable the correct classification of the subsets of data not covered by the final classifier.

apply a linear layer to classify the samples as in a final classifier. We evaluate the final task-agnostic accuracy across different tasks and average results for selected classifiers, as shown in Figure 2.

We observe that the average accuracy of the penultimate classifier matches or even surpasses that of the final classifier on the older tasks data. In the case of naive finetuning, for all tasks aside from the last one intermediate classifiers achieve the highest accuracy. For exemplar-free LwF there is no clear pattern, but intermediate classifiers also outperform the final classifier on many tasks. In the case of exemplar-based methods such as FT+Ex and BiC, the performance on each task more or less steadily improves with the deeper classifiers, but the deepest two intermediate classifiers show comparable performance to the final one. Overall, our results further confirm the potential benefits of auxiliary intermediate classifiers in continual learning scenarios.

## 2.3 DIVERSITY OF THE AUXILIARY CLASSIFIER PREDICTIONS

Our previous analysis suggests that intermediate representations in the network exhibit higher stability than and can be used for classification in continual learning with performance comparable to the final classifier, at least in the deeper classifiers. Due to the better stability of the representations, in exemplar-free scenarios, the classifiers built on top of intermediate representations might significantly outperform the final classifier when evaluated in isolation. When using exemplars, later classifiers usually perform better than the early ones on all tasks, but our previous experiment did not verify if those classifiers learn to cover the same subsets of data, or if they learn to operate differently from each other. In the context of continual learning, multiple classifiers could forget and remember different sets of data, which could be leveraged to improve the overall performance.

To investigate the diversity among the auxiliary classifiers, we measure *unique accuracy* - the percentage of the samples that are correctly predicted only by this classifier. If a given classifier has 10% unique accuracy, it means that 10% of all task data is correctly classified by only this classifier and misclassified by all other classifiers. We present the results in Figure 3. The intermediate classifiers learn to specialize to some degree, especially on the older tasks. The trend is again more visible for the naive and exemplar-free settings, but it occurs for all analyzed methods, and all intermediate classifiers exhibit some degree of unique accuracy. This means that attaching auxiliary classifiers can enhance the network with the knowledge that it cannot learn in a standard process, potentially enabling better classification in continual learning settings.

## 2.4 IMPROVING ACs PERFORMANCE THROUGH GRADIENT PROPAGATION

The results presented in the previous section indicate that auxiliary classifiers (ACs) learned through linear probing could be utilized to greatly improve performance in continual learning. However, we hypothesize that the performance of the classifiers would further improve if trained with enabled gradient propagation from classifiers through the network. To verify our hypothesis, we jointly train the same network with 6 ACs with enabled gradient propagation and plot in Figure 4 the difference between the final average accuracy for each classifier in comparison to linear probing.

Training classifiers together with the network generally improves the performance of early and middle classifiers. While the performance of later classifiers slightly degrades for FT and to a degree for LwF, in the other settings we observe significant accuracy gains for the intermediate layers and no degradation in the deeper ones. We hypothesize that higher gains in the exemplar-based settings can be attributed to the fact that the networks can better retain the knowledge during training, which is consistent with our findings from Section 2.1 where those settings exhibit higher stability.

In Appendix C, we perform the experiments from Sections 2.1 to 2.3 for networks with ACs trained together with the main network and demonstrate that our previous observations also hold in this setup. As the classifiers trained jointly with the backbone network with enabled gradient propagation demonstrate better accuracy, in all later stages of our work we use this setup.

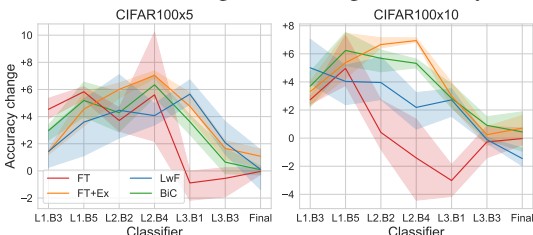

Figure 4: Accuracy changes with full training.

## 3 ENHANCING CONTINUAL LEARNING WITH AUXILIARY CLASSIFIERS

### 3.1 COMBINING PREDICTIONS FROM MULTI-CLASSIFIER NETWORKS.

Our analysis in Section 2 demonstrates that auxiliary classifiers (ACs) can learn to classify different subsets of data than just a standard, single-classifier network, which hints that combining their predictions should yield improved accuracy. Therefore, we advocate the use of such multi-classifier networks in continual learning. Formally, we consider a neural network composed of backbone $f = f_N(...(f_1(x)))$ and final classifier $g$, where $f_1, ..., f_N$ are submodules in the backbone. The standard network prediction $y$ for a given input $x$ can be written as $y = g(f(x))$. We introduce additional $N - 1$ auxiliary classifiers $\hat{g}_i$ on top of the backbone sub-modules $f_1, f_2, ..., f_{N-1}$. During inference with such multi-classifier network, we obtain $N$ predictions: $y_1 = \hat{g}_1(f_1(x)), y_2 = \hat{g}_2(f_2(x)), ..., y_{N-1} = \hat{g_{N-1}}(f_{N-1}(x)), y_N = g(f(x))$ and select the prediction $y_i$ where the class predicted by the corresponding probability distribution $p_i$ has maximum confidence. Therefore, we return $y = y_{\arg\max_{i \in \{1,...,N\}} \max_k p_i^{(k)}}$, where $p_i^{(k)}$ represents the predicted probability for class $k$ in the distribution $p_i$. We refer to this simple inference paradigm as *static inference* and use it in most of the experiments, as we find it performs well across all tested settings.

Inspired by the early-exit models (Panda et al., 2016; Teerapittayanon et al., 2016; Kaya et al., 2019), we also consider using ACs as a means to reduce the average computational cost of the classification through *dynamic inference*. Specifically, in this scenario, we perform inference sequentially through the classifiers $\hat{g}_1, \hat{g}_2, ..., g$, and at each stage $i$, we compute the probability distribution $p_i$ corresponding to the prediction of $i$-th classifier. If the confidence exceeds a set threshold $\lambda$, we return the corresponding prediction $y_i$. If no prediction satisfies the threshold, we use the static inference rule to determine the prediction. Formally, we define this process as:

$$y = \begin{cases} y_{\min\{i \in \{1,...,N\} | \max_k p_i^{(k)} \geq \lambda\}} & \text{if such } i \text{ exists,} \\ y_{\arg\max_{i \in \{1,...,N\}} \max_k p_i^{(k)}} & \text{otherwise.} \end{cases} \quad (1)$$

By varying the confidence threshold, one can trade off the amount of computation performed by the network for slightly lower performance, which allows such a model to be deployed in settings requiring computational adaptability.

Note that our use of ACs is different from the early-exit literature, where the model accuracy usually monotonically improves when going through subsequent classifiers and the model returns the prediction of the last classifier in case no classifier can satisfy the exit threshold. As we already demonstrated in Section 2, in continual learning the accuracy and quality of intermediate predictions significantly vary for different tasks, and the last classifier is not always the best one for a given subset of data. Please refer to Appendix E for a comparison between the performance of the standard early-exit inference rule and our method of using the ACs.

Table 1: Final accuracy of several continual learning methods on CIFAR100 and ImageNet100 benchmarks before and after enhanced with auxiliary classifiers (ACs). Adding ACs improves the performance of all methods across all the benchmarks, demonstrating the robustness of our idea.

| Method | FT | FT+Ex | GDumb | ANCL | BiC | DER++ | ER | EWC | LwF | LODE | SSIL | Avg |
|---|---|---|---|---|---|---|---|---|---|---|---|---|
| | | | | | CIFAR100x5 | | | | | | | |
| Base | 18.68$_{\pm0.31}$ | 38.35$_{\pm0.86}$ | 19.09$_{\pm0.44}$ | 37.71$_{\pm1.14}$ | 47.66$_{\pm0.43}$ | 36.54$_{\pm4.62}$ | 34.55$_{\pm0.21}$ | 18.95$_{\pm0.29}$ | 38.26$_{\pm0.98}$ | 42.82$_{\pm0.84}$ | 45.62$_{\pm0.16}$ | 34.38$_{\pm0.66}$ |
| +AC | **28.18**$_{\pm1.07}$ | **38.75**$_{\pm0.26}$ | **23.29**$_{\pm0.54}$ | **39.83**$_{\pm1.22}$ | **50.40**$_{\pm0.68}$ | **42.37**$_{\pm3.27}$ | **39.77**$_{\pm0.32}$ | **28.96**$_{\pm1.13}$ | **40.55**$_{\pm0.95}$ | **49.13**$_{\pm0.35}$ | **48.35**$_{\pm0.50}$ | **39.05**$_{\pm0.83}$ |
| Δ | +9.49$_{\pm0.96}$ | +0.39$_{\pm0.90}$ | +4.20$_{\pm0.16}$ | +2.12$_{\pm1.03}$ | +2.74$_{\pm0.83}$ | +5.83$_{\pm1.97}$ | +5.22$_{\pm0.38}$ | +10.02$_{\pm1.39}$ | +2.29$_{\pm0.25}$ | +6.31$_{\pm0.81}$ | +2.72$_{\pm0.42}$ | +4.67$_{\pm0.47}$ |
| | | | | | CIFAR100x10 | | | | | | | |
| Base | 10.27$_{\pm0.05}$ | 34.51$_{\pm0.40}$ | 22.22$_{\pm0.72}$ | 30.69$_{\pm0.62}$ | 42.87$_{\pm1.51}$ | 38.54$_{\pm0.65}$ | 32.31$_{\pm0.82}$ | 10.20$_{\pm0.35}$ | 29.56$_{\pm0.44}$ | 38.87$_{\pm0.45}$ | 42.29$_{\pm0.49}$ | 30.21$_{\pm0.18}$ |
| +AC | **16.88**$_{\pm1.08}$ | **36.97**$_{\pm0.39}$ | **27.74**$_{\pm0.73}$ | **31.37**$_{\pm0.94}$ | **46.19**$_{\pm1.47}$ | **39.64**$_{\pm1.00}$ | **37.32**$_{\pm0.28}$ | **19.12**$_{\pm0.88}$ | **30.31**$_{\pm1.14}$ | **45.67**$_{\pm0.52}$ | **44.17**$_{\pm0.28}$ | **34.13**$_{\pm0.24}$ |
| Δ | +6.62$_{\pm1.06}$ | +2.46$_{\pm0.31}$ | +5.52$_{\pm1.13}$ | +0.68$_{\pm0.79}$ | +3.31$_{\pm2.62}$ | +1.10$_{\pm1.08}$ | +5.01$_{\pm0.98}$ | +8.92$_{\pm1.06}$ | +0.74$_{\pm0.91}$ | +6.80$_{\pm0.93}$ | +1.88$_{\pm0.77}$ | +3.91$_{\pm0.41}$ |
| | | | | | ImageNet100x5 | | | | | | | |
| Base | 23.27$_{\pm0.39}$ | 44.05$_{\pm0.69}$ | 21.29$_{\pm0.59}$ | 60.79$_{\pm0.06}$ | 62.55$_{\pm0.53}$ | 40.39$_{\pm6.07}$ | 38.65$_{\pm0.43}$ | 23.36$_{\pm0.64}$ | 59.60$_{\pm0.27}$ | 49.88$_{\pm0.56}$ | 60.54$_{\pm0.32}$ | 44.03$_{\pm0.48}$ |
| +AC | **34.93**$_{\pm0.65}$ | **46.75**$_{\pm0.61}$ | **25.30**$_{\pm1.14}$ | **62.99**$_{\pm0.30}$ | **65.22**$_{\pm0.27}$ | **54.65**$_{\pm0.37}$ | **44.46**$_{\pm0.47}$ | **35.09**$_{\pm0.17}$ | **61.07**$_{\pm0.57}$ | **56.23**$_{\pm0.66}$ | **63.89**$_{\pm0.18}$ | **50.05**$_{\pm0.10}$ |
| Δ | +11.67$_{\pm0.77}$ | +2.71$_{\pm0.85}$ | +4.01$_{\pm0.61}$ | +2.21$_{\pm0.35}$ | +2.67$_{\pm0.79}$ | +14.26$_{\pm6.25}$ | +5.81$_{\pm0.45}$ | +11.73$_{\pm0.71}$ | +1.47$_{\pm0.45}$ | +6.35$_{\pm1.22}$ | +3.35$_{\pm0.48}$ | +6.02$_{\pm0.50}$ |
| | | | | | ImageNet100x10 | | | | | | | |
| Base | 14.40$_{\pm0.30}$ | 35.94$_{\pm0.86}$ | 22.55$_{\pm0.62}$ | 49.96$_{\pm0.46}$ | 56.32$_{\pm0.47}$ | 31.49$_{\pm10.1}$ | 32.45$_{\pm0.35}$ | 14.69$_{\pm0.20}$ | 49.15$_{\pm0.38}$ | 45.75$_{\pm0.50}$ | 56.35$_{\pm0.51}$ | 37.19$_{\pm0.77}$ |
| +AC | **22.14**$_{\pm0.16}$ | **39.26**$_{\pm0.61}$ | **25.93**$_{\pm0.52}$ | **52.07**$_{\pm0.50}$ | **57.23**$_{\pm0.87}$ | **43.05**$_{\pm4.19}$ | **37.10**$_{\pm1.20}$ | **23.25**$_{\pm0.55}$ | **49.51**$_{\pm0.71}$ | **51.39**$_{\pm0.91}$ | **57.71**$_{\pm0.08}$ | **41.69**$_{\pm0.54}$ |
| Δ | +7.74$_{\pm0.37}$ | +3.32$_{\pm0.90}$ | +3.38$_{\pm0.37}$ | +2.11$_{\pm0.32}$ | +0.91$_{\pm0.42}$ | +11.56$_{\pm12.75}$ | +4.65$_{\pm0.88}$ | +8.56$_{\pm0.39}$ | +0.36$_{\pm1.05}$ | +5.64$_{\pm0.90}$ | +1.35$_{\pm0.59}$ | +4.51$_{\pm1.19}$ |

## 3.2 AC-ENHANCED CONTINUAL LEARNING METHODS.

To demonstrate the effectiveness of our idea, we extend several continual learning methods with auxiliary classifiers (ACs) and examine their performance. In total, we investigate ACs with the following continual learning methods: FT (Masana et al., 2022), GDUMB (Prabhu et al., 2020), EWC (Kirkpatrick et al., 2017), LwF (Li & Hoiem, 2017), ER (Riemer et al., 2018), DER++ (Buzzega et al., 2020), BiC (Wu et al., 2019), SSIL (Ahn et al., 2021), ANCL (Kim et al., 2023) and LODE (Liang & Li, 2024). FT (finetuning) is a naive scenario where the network is trained without any additional continual learning loss, and the stability can only be enforced through the additional use of exemplars (FT+Ex, where we simply mix the exemplars with the training data from the new task and do not balance the training batches). EWC and LwF improve the stability of the network through additional regularization loss that penalizes change either to model weights or activations. We use both methods without exemplars, as Masana et al. (2022) shows they do not improve from replay. ER uses replay with balanced memory batches, with each batch containing the same amount of old and new samples. DER++ extends this idea by adding the replay on logits. BiC and SSIL also employ distillation and replay, but provide additional mechanisms to counter task recency bias. ANCL uses knowledge distillation from two networks, a 'stable' one as in LwF and the 'plastic' one overfitted to a new task. LODE uses replay and disentangles the training loss between stability and plasticity terms to reduce forgetting.

For all the methods, we replicate the method logic (loss) across all the classifiers and do not introduce classifier-specific parameters. If the original method introduces a hyperparameter, we use the same value for this hyperparameter across all the classifiers. We also use the same batches of data for each classifier during the training. Similar to Kaya et al. (2019), to prevent overfitting the network to the early layer classifiers we scale the total loss of each classifier according to its position so that the losses from early classifiers are weighted less than the losses for the final classifier.

## 4 EXPERIMENTAL RESULTS

In this section, we show the main results for AC-enhanced networks on standard continual learning benchmarks. We use FACIL Masana et al. (2022) framework and conduct the experiments on CIFAR100 (Krizhevsky, 2009) and ImageNet100 Deng et al. (2009) (the first 100 classes from ImageNet) splits into tasks containing different classes. We use ResNet32 for experiments on CIFAR100 and ResNet18 He et al. (2016) for experiments on ImageNet100 and add 6 ACs for the main experiments in both settings. For ResNet32, we follow the previously described AC placement, and for ResNet18 we attach the AC to all residual blocks, excluding the first and last one followed by the final classifier. For all exemplar-based methods (BiC, DER++, ER, GDUMB, LODE and SSIL) we use a fixed-size memory budget of 2000 exemplars updated after each task. We report the results averaged over 3 random seeds. Refer to Appendix I for more details.

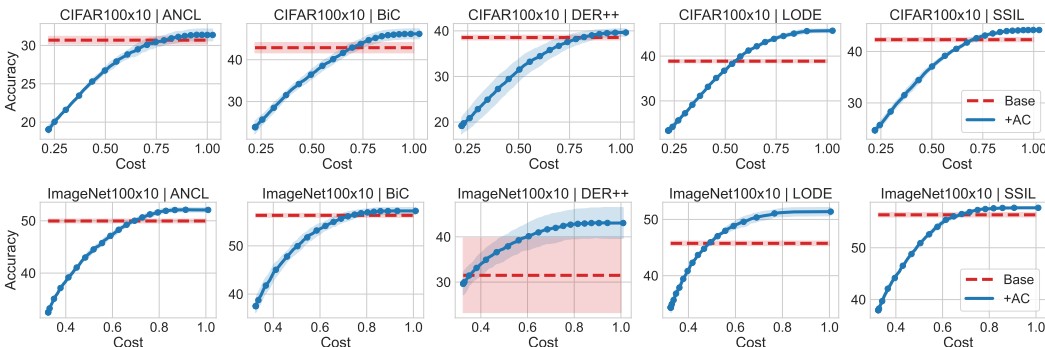

Figure 5: Dynamic inference plots for several continual learning methods extended with auxiliary classifiers compared with the baselines for CIFAR100 (top row) and ImageNet100 (bottom row) split into 10 tasks. Adding auxiliary classifiers not only improves the performance but also can be used to reduce the computational cost of the inference across all the methods. We report cost in FLOPs relative to the non-AC version of the method. We evaluate dynamic inference using $\lambda \in \{0.01, 0.02, ..., 0.99, 1.00\}$ and mark every 5% confidence threshold with the dots.

**Classic continual learning benchmarks.** We present our main results on CIFAR100 and ImageNet100 split into 5 and 10 disjoint, equally sized tasks in Table 1. Adding auxiliary classifiers improves the final performance across all methods and settings, with the average relative improvement over all tested methods exceeding 10% of the baseline accuracy in all tested scenarios. Naive methods such as FT and EWC improve significantly, and exemplar-based methods (BiC, LODE, SSIL) usually achieve a bigger boost from the addition of auxiliary classifiers compared to distillation-based ANCL and LwF. We also observe slightly better improvements on ImageNet100 as compared to CIFAR100, which we attribute to the better network capacity that enables more expressivity in the intermediate representations. In Appendix F, we also present the results with longer, 20, and 50 task sequences, where our method likewise outperforms the baselines. The results prove the robustness of our idea, even though our utilization of auxiliary classifiers is motivated by simplicity and we did not optimize AC placement, architecture, or training beyond the simple well-known recipes.

**Reducing the network computation through auxiliary classifiers.** Our results in Section 4 demonstrate that enhancing continual learning methods with auxiliary classifiers results in improved final performance at the full computational budget of the network. In this section, we instead investigate dynamic inference described in Section 3 as a means to accelerate network inference. Namely, we evaluate the performance of selected continual learning methods on CIFAR100 and ImageNet100 split into 10 tasks using a dense grid of confidence thresholds ($\lambda \in \{0.01, 0.02, ..., 0.99, 1.00\}$) and measure the average FLOPS per sample relative to the cost of using a standard, single-classifier method. We plot resulting cost-accuracy characteristics in comparison with the standard counterparts' performance in Figure 5. In Appendix G.2, we also provide dynamic inference plots for the setting analyzed in this section. Using ACs and dynamic inference, we are able to match the performance of single-classifier methods using approximately 50%-70% of their computation on CIFAR100 and approximately 40%-70% of the computation on ImageNet100. Interestingly, for most methods, performance seems to saturate at 80%-90% of compute, which means we can potentially save this much computation without any accuracy decrease. Similar to the previous section, the improvement on ImageNet is slightly better, which we attribute to the better capacity of ResNet18 used in this setting. Dynamic inference is fairly robust to the thresholding, with any confidence thresholds above 75% still outperforming the baseline and thresholds above 90% achieving close to no degradation in performance in all settings.

**ACs in warm-start continual learning.** A common scenario in continual learning is warm start (Magistri et al.; Goswami et al., 2024) that simulates starting from a pre-trained model checkpoint. In this scenario, the model is trained continually, but the first task contains a large portion of the whole data so that during this task the network can already accumulate a lot of knowledge, as in the case of pre-training. Such a scenario is an interesting study for continual learning due to the practical benefits of using pre-trained models and the difference in learning dynamics when starting

Table 2: Adding auxiliary classifiers (ACs) is beneficial to the final network accuracy when starting from a pre-trained state, which we simulate by using CIFAR splits with 50 classes in the first task.

| Method | FT | FT+Ex | GDumb | ANCL | BiC | ER | EWC | LwF | LODE | SSIL | Avg |
|---|---|---|---|---|---|---|---|---|---|---|---|
| | | | | | CIFAR100x6 | | | | | | |
| Base | $16.18_{\pm0.65}$ | $\mathbf{40.38}_{\pm0.75}$ | $17.38_{\pm0.33}$ | $42.85_{\pm1.07}$ | $45.87_{\pm2.53}$ | $\mathbf{38.11}_{\pm0.10}$ | $17.08_{\pm1.11}$ | $42.72_{\pm0.60}$ | $42.28_{\pm0.46}$ | $46.78_{\pm0.15}$ | $34.96_{\pm0.41}$ |
| +AC | $\mathbf{22.37}_{\pm1.28}$ | $38.12_{\pm0.77}$ | $\mathbf{22.60}_{\pm0.31}$ | $\mathbf{43.97}_{\pm0.41}$ | $\mathbf{48.99}_{\pm0.80}$ | $37.81_{\pm0.58}$ | $\mathbf{25.49}_{\pm0.97}$ | $\mathbf{43.45}_{\pm0.74}$ | $\mathbf{44.95}_{\pm0.28}$ | $\mathbf{48.97}_{\pm0.28}$ | $\mathbf{37.67}_{\pm0.30}$ |
| Δ | $+6.19_{\pm1.72}$ | $-2.26_{\pm0.35}$ | $+5.22_{\pm0.19}$ | $+1.12_{\pm1.35}$ | $+3.11_{\pm3.29}$ | $-0.30_{\pm0.68}$ | $+8.40_{\pm0.71}$ | $+0.72_{\pm1.13}$ | $+2.67_{\pm0.28}$ | $+2.19_{\pm0.36}$ | $+2.71_{\pm0.32}$ |
| | | | | | CIFAR100x11 | | | | | | |
| Base | $7.90_{\pm0.30}$ | $36.41_{\pm1.06}$ | $16.55_{\pm0.41}$ | $33.86_{\pm0.11}$ | $42.38_{\pm0.64}$ | $34.86_{\pm0.56}$ | $8.01_{\pm0.88}$ | $32.13_{\pm0.72}$ | $38.17_{\pm0.17}$ | $41.46_{\pm0.84}$ | $29.17_{\pm0.05}$ |
| +AC | $\mathbf{11.91}_{\pm1.59}$ | $\mathbf{36.80}_{\pm0.45}$ | $\mathbf{22.73}_{\pm0.74}$ | $\mathbf{34.94}_{\pm0.95}$ | $\mathbf{45.37}_{\pm0.44}$ | $\mathbf{36.80}_{\pm0.53}$ | $\mathbf{16.05}_{\pm0.96}$ | $\mathbf{35.31}_{\pm1.42}$ | $\mathbf{40.97}_{\pm0.22}$ | $\mathbf{45.70}_{\pm0.59}$ | $\mathbf{32.66}_{\pm0.35}$ |
| Δ | $+4.00_{\pm1.48}$ | $+0.39_{\pm0.63}$ | $+6.17_{\pm0.38}$ | $+1.08_{\pm0.85}$ | $+2.99_{\pm0.21}$ | $+1.94_{\pm1.07}$ | $+8.04_{\pm0.67}$ | $+3.18_{\pm0.77}$ | $+2.80_{\pm0.35}$ | $+4.24_{\pm1.10}$ | $+3.48_{\pm0.31}$ |

from a trained model. To validate how our model behaves in a warm start scenario, we train the methods from the previous sections on CIFAR100 and use 50 classes for the first task to simulate a pre-training phase. After the first task, we split the remaining classes evenly into 5 or 10 additional tasks (we refer to both settings as 6 and 11 task split). Aside from the task split, we perform the experiments as described in the previous sections and report the results in Table 2. We observe that the performance of the methods enhanced with the ACs generally improves, aside from ER and FT+Ex on 6 tasks; however, those methods are quite naive and ACs work for them in the other settings. Overall, we can also conclude that our idea is almost universally beneficial in warm start setting.

**Number of ACs.** Our approach requires deciding the AC placement, which will affect the performance. To test the robustness of our idea, we perform an ablation where we change the number of classifiers, using either half of them or twice as much (we either drop every other classifier in our standard setting or attach an additional one to the ResNet blocks in between the previously selected classifiers). We measure the improvement obtained upon the baseline (already reported in Table 1) and the additional computational cost incurred by the ACs when using 3, 6 (standard setting), and 12 ACs and report it in Table 3. While the best setup across several continual learning methods varies, the number of ACs does not significantly affect the accuracy and their addition does not significantly increase the computation in the network. In all cases, the networks with auxiliary classifiers achieve an improvement upon the baseline, underlining the robustness of our idea.

Table 3: Difference w.r.t. baseline single-classifier methods when using a different number of auxiliary classifiers (ACs). ACs robustly improve the final accuracy of continual learning methods, regardless of the number of classifiers used.

| | FLOPS | FT | FT+Ex | GDumb | ANCL | BiC | ER | EWC | LwF | LODE | SSIL | Avg |
|---|---|---|---|---|---|---|---|---|---|---|---|---|
| NoAC | 69.90M (1x) | | | | | | | | | | | |
| | | | | | | CIFAR100x5 | | | | | | |
| 3AC | 70.72M (1.01x) | $+7.61_{\pm0.68}$ | $+0.70_{\pm0.83}$ | $+5.08_{\pm0.87}$ | $+2.14_{\pm1.05}$ | $+2.62_{\pm0.60}$ | $+4.63_{\pm0.38}$ | $+8.94_{\pm0.40}$ | $+0.95_{\pm1.18}$ | $+4.19_{\pm0.63}$ | $+2.65_{\pm0.58}$ | $+3.95_{\pm0.39}$ |
| 6AC | 71.55M (1.02x) | $+9.49_{\pm0.96}$ | $+0.39_{\pm0.90}$ | $+4.20_{\pm0.16}$ | $+2.12_{\pm1.03}$ | $+2.74_{\pm0.83}$ | $+5.22_{\pm0.38}$ | $+10.02_{\pm1.39}$ | $+2.29_{\pm0.25}$ | $+6.31_{\pm0.81}$ | $+2.72_{\pm0.42}$ | $+4.55_{\pm0.43}$ |
| 12AC | 72.97M (1.04x) | $+9.09_{\pm0.81}$ | $+1.67_{\pm1.28}$ | $+5.15_{\pm0.08}$ | $+2.45_{\pm0.74}$ | $+3.57_{\pm0.47}$ | $+4.46_{\pm0.54}$ | $+9.97_{\pm0.35}$ | $+2.75_{\pm1.26}$ | $+5.43_{\pm0.65}$ | $+2.92_{\pm0.53}$ | $+4.74_{\pm0.20}$ |
| | | | | | | CIFAR100x10 | | | | | | |
| 3AC | 70.84M (1.01x) | $+6.48_{\pm0.43}$ | $+3.05_{\pm0.99}$ | $+5.74_{\pm0.47}$ | $+1.71_{\pm0.31}$ | $+2.85_{\pm2.19}$ | $+4.44_{\pm1.00}$ | $+7.92_{\pm0.61}$ | $+0.53_{\pm0.44}$ | $+6.13_{\pm0.23}$ | $+2.02_{\pm1.69}$ | $+4.09_{\pm0.42}$ |
| 6AC | 71.77M (1.03x) | $+6.62_{\pm1.06}$ | $+2.46_{\pm0.31}$ | $+5.52_{\pm1.13}$ | $+0.68_{\pm0.79}$ | $+3.31_{\pm2.62}$ | $+5.01_{\pm0.98}$ | $+8.92_{\pm1.06}$ | $+0.74_{\pm0.91}$ | $+6.80_{\pm0.93}$ | $+1.88_{\pm0.77}$ | $+4.20_{\pm0.47}$ |
| 12AC | 73.36M (1.05x) | $+4.63_{\pm1.46}$ | $+2.59_{\pm1.19}$ | $+6.12_{\pm0.78}$ | $+1.85_{\pm1.49}$ | $+3.98_{\pm2.01}$ | $+4.95_{\pm1.05}$ | $+6.57_{\pm0.97}$ | $+1.14_{\pm0.38}$ | $+6.68_{\pm0.79}$ | $+1.98_{\pm0.91}$ | $+4.05_{\pm0.60}$ |

**Compatibility with Vision Transformers.** In addition to our architectural study in Section 4, we also explore the use of Vision Transformer (ViT) architecture (Dosovitskiy, 2020) enhanced with ACs. We train the ViT-base model from scratch on ImageNet100 split into 10 tasks (with results for 5 tasks in Appendix G.6), with additional classifiers attached to each transformer block (11 ACs in total). While training from scratch is not a usual setup for ViTs, we consider our comparison fair, as we use the same setup for baseline and AC-enhanced methods. We plot dynamic inference results for ViT with ACs compared with baseline in Figure 6, using a subset of methods in this setting due to computation constraints. Overall, we see that ACs also perform well for transformer models. Transformer architecture is also more suited to enhancements with ACs, as they also allow better computational savings and induce significantly less overhead.

**Deeper convolutional models.** To test our method with deeper convolutional models, we evaluate it with 19-layer VGG19 network (Simonyan & Zisserman, 2014) on CIFAR100 split into 5 and

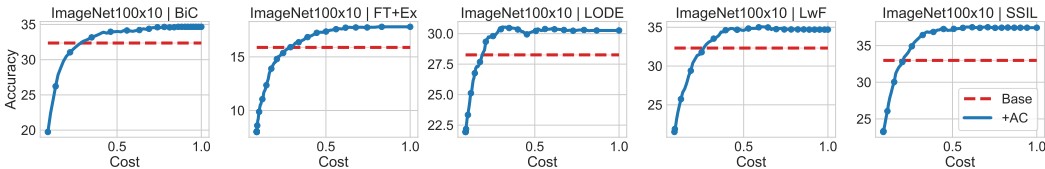

Figure 6: Dynamic inference plots for continual learning methods extended with auxiliary classifiers and the baselines with Vision Transformers trained from scratch on ImageNet100.

10 tasks. We keep the training setup from Section 4, and compare three AC-enhanced variants of each method with its base version. We use the same AC architecture as in the ResNet experiments and attach the AC either to every each of 18 intermediate layer outputs, every other convolutional layer and both fully connected intermediate layers (10 ACs), or every 4th convolutional layer and both fully connected intermediate layers (6 ACs). We summarize the results of our experiments in Table 4 and provide matching dynamic inference plots for those experiments in Appendix G.5. All AC setups outperform the baseline methods, with more ACs usually performing slightly better.

Table 4: CIFAR100 results for VGG19 network enhanced with different number of ACs.

| Method | FT | FT+Ex | GDumb | ANCL | BiC | DER++ | ER | EWC | LwF | LODE | SSIL | Avg |
|---|---|---|---|---|---|---|---|---|---|---|---|---|
| | | | | | CIFAR100x5 | | | | | | | |
| Base | $9.52_{\pm0.17}$ | $34.20_{\pm0.48}$ | $28.79_{\pm0.66}$ | $19.50_{\pm0.97}$ | $44.30_{\pm1.70}$ | $41.88_{\pm0.44}$ | $28.85_{\pm0.83}$ | $9.37_{\pm0.43}$ | $21.04_{\pm0.79}$ | $40.08_{\pm0.52}$ | $42.49_{\pm0.73}$ | $29.09_{\pm0.18}$ |
| +6AC | $16.73_{\pm0.31}$ | $35.29_{\pm0.39}$ | $30.99_{\pm0.69}$ | $26.96_{\pm1.00}$ | $50.36_{\pm0.73}$ | $45.02_{\pm0.13}$ | $32.48_{\pm0.48}$ | $17.16_{\pm0.37}$ | $28.80_{\pm0.97}$ | $45.67_{\pm0.39}$ | $47.39_{\pm0.30}$ | $34.26_{\pm0.06}$ |
| +10AC | $18.91_{\pm0.36}$ | $36.99_{\pm0.15}$ | $31.55_{\pm0.35}$ | $29.73_{\pm0.46}$ | $52.69_{\pm0.56}$ | $43.94_{\pm0.58}$ | $33.95_{\pm0.41}$ | $19.78_{\pm0.32}$ | $31.28_{\pm0.73}$ | $46.27_{\pm0.58}$ | $48.29_{\pm1.02}$ | $35.76_{\pm0.15}$ |
| +18AC | $21.16_{\pm0.13}$ | $37.54_{\pm0.19}$ | $31.63_{\pm0.78}$ | $32.61_{\pm0.60}$ | $52.56_{\pm0.53}$ | $43.94_{\pm0.82}$ | $34.86_{\pm0.42}$ | $20.54_{\pm0.28}$ | $32.54_{\pm0.22}$ | $47.62_{\pm0.14}$ | $49.68_{\pm0.10}$ | $36.79_{\pm0.10}$ |
| | | | | | CIFAR100x10 | | | | | | | |
| Base | $18.91_{\pm0.14}$ | $42.56_{\pm0.55}$ | $26.64_{\pm1.24}$ | $40.73_{\pm0.57}$ | $52.75_{\pm0.70}$ | $45.52_{\pm0.29}$ | $33.82_{\pm0.20}$ | $18.72_{\pm0.36}$ | $39.54_{\pm0.59}$ | $46.69_{\pm0.64}$ | $47.79_{\pm0.11}$ | $37.61_{\pm0.19}$ |
| +6AC | $26.71_{\pm0.62}$ | $42.78_{\pm0.62}$ | $29.61_{\pm1.11}$ | $47.64_{\pm0.66}$ | $56.62_{\pm1.13}$ | $51.16_{\pm0.64}$ | $37.45_{\pm0.40}$ | $26.98_{\pm0.68}$ | $44.81_{\pm0.64}$ | $52.03_{\pm0.07}$ | $52.87_{\pm0.42}$ | $42.61_{\pm0.43}$ |
| +10AC | $29.16_{\pm0.23}$ | $43.05_{\pm0.45}$ | $31.36_{\pm0.73}$ | $49.12_{\pm0.70}$ | $58.05_{\pm0.44}$ | $51.03_{\pm0.23}$ | $39.06_{\pm0.70}$ | $29.40_{\pm0.32}$ | $46.51_{\pm0.52}$ | $50.39_{\pm0.65}$ | $55.30_{\pm0.32}$ | $43.86_{\pm0.22}$ |
| +18AC | $31.47_{\pm0.34}$ | $43.53_{\pm0.39}$ | $31.06_{\pm0.82}$ | $48.49_{\pm1.02}$ | $59.03_{\pm0.41}$ | $50.67_{\pm0.89}$ | $39.91_{\pm0.33}$ | $30.66_{\pm0.63}$ | $48.22_{\pm0.16}$ | $51.27_{\pm0.85}$ | $56.35_{\pm0.16}$ | $44.61_{\pm0.23}$ |

**Alternative AC architectures.** In our main work, we investigate a simple setup with independent classifiers. Early-exit works such as (Wójcik et al., 2023) propose more complex dynamic architectures, where subsequent classifiers are connected and their predictions are combined through a weighted ensemble. Those architectures induce only a slight parameter and computation overhead, but in a standard supervised learning setting can improve the performance of intermediate classifiers through sharing the knowledge between them. We investigate those architectures in continual learning on the set of methods analyzed in previous sections on split CIFAR100 benchmarks and present the results in Table 5. Similar to the AC density ablation, we do not observe a clear improvement from changing the setup. We hypothesize that connecting the classifiers makes them no longer independent, which negates the benefits yielded in continual learning by the classifier diversity.

Table 5: Difference w.r.t. baseline single-classifier methods when using a different auxiliary classifier architecture: cascading (C) and ensebling (E) from Wójcik et al. (2023). Similar to Table 3, ACs universally improve the accuracy with small differences in performance between the architectures.

| Method | FT | FT+Ex | GDumb | ANCL | BiC | ER | EWC | LwF | LODE | SSIL | Avg |
|---|---|---|---|---|---|---|---|---|---|---|---|
| | | | | | CIFAR100x5 | | | | | | |
| AC | $+5.70_{\pm5.24}$ | $+0.24_{\pm0.67}$ | $+2.52_{\pm2.30}$ | $+1.27_{\pm1.37}$ | $+1.65_{\pm1.61}$ | $+3.13_{\pm2.87}$ | $+6.01_{\pm5.57}$ | $+1.37_{\pm1.27}$ | $+3.79_{\pm3.50}$ | $+1.63_{\pm1.52}$ | $+2.73_{\pm2.51}$ |
| AC+C | $+5.41_{\pm4.99}$ | $-0.72_{\pm0.70}$ | $+2.49_{\pm2.27}$ | $+1.20_{\pm1.37}$ | $+1.16_{\pm1.06}$ | $+2.57_{\pm2.37}$ | $+6.16_{\pm5.62}$ | $+0.66_{\pm0.77}$ | $+2.89_{\pm2.66}$ | $+1.26_{\pm1.23}$ | $+2.31_{\pm2.12}$ |
| AC+E | $+6.47_{\pm5.91}$ | $+0.38_{\pm1.03}$ | $+2.00_{\pm1.83}$ | $+1.28_{\pm39.62}$ | $+1.30_{\pm1.20}$ | $+2.32_{\pm2.16}$ | $+6.58_{\pm6.01}$ | $+1.14_{\pm1.18}$ | $+2.79_{\pm2.61}$ | $+1.29_{\pm1.23}$ | $+2.56_{\pm2.11}$ |
| | | | | | CIFAR100x10 | | | | | | |
| AC | $+6.62_{\pm1.06}$ | $+2.46_{\pm0.31}$ | $+5.52_{\pm1.13}$ | $+0.68_{\pm0.79}$ | $+3.31_{\pm2.62}$ | $+5.01_{\pm0.98}$ | $+8.92_{\pm1.06}$ | $+0.74_{\pm0.91}$ | $+6.80_{\pm0.93}$ | $+1.88_{\pm0.77}$ | $+4.20_{\pm0.47}$ |
| AC+C | $+6.98_{\pm1.05}$ | $+2.63_{\pm0.92}$ | $+5.56_{\pm0.96}$ | $+1.60_{\pm0.70}$ | $+3.63_{\pm1.51}$ | $+5.19_{\pm1.08}$ | $+8.35_{\pm0.39}$ | $+1.27_{\pm0.47}$ | $+5.45_{\pm0.17}$ | $+2.53_{\pm0.56}$ | $+4.32_{\pm0.16}$ |
| AC+E | $+7.35_{\pm0.14}$ | $+3.27_{\pm0.74}$ | $+5.09_{\pm0.40}$ | $+2.28_{\pm0.21}$ | $+3.68_{\pm2.16}$ | $+4.77_{\pm0.52}$ | $+8.62_{\pm1.12}$ | $+1.18_{\pm0.30}$ | $+5.85_{\pm0.68}$ | $+1.53_{\pm0.67}$ | $+4.36_{\pm0.33}$ |

## 5 COMPUTATIONAL OVERHEAD FROM THE INTRODUCTION OF ACs

ACs require extra memory, and in principle make training more complex with additional network modules and hyperparameters. We focus on an offline class-incremental setting, so we consider the

inference time advantages of our method - increased performance and the ability to reduce network computation - to be more important. However, to provide a fair overview of our approach, we show parameter and memory overhead for the network setups tested in Section 4 in Table 6, alongside the training times for the runs on CIFAR100 with different numbers of ACs in Table 7.

For small ResNet models, the memory and parameter overhead from ACs is significant, but it becomes significantly lower for larger and deeper models such as VGG19 and ViT-base. In the case of ResNet models, they are very parameter efficient, so ACs induce visible overhead. **However, it does not directly translate to significantly higher computation, as shown in FLOPs in dynamic inference plots.**

Table 6: Parameters and inference memory usage in the base and AC-enhanced models.

|  | Par(base) | Par(AC) | Mem(base) | Mem(AC) |
|---|---|---|---|---|
| ResNet32(+3AC) | 0.47M | 1.19M | 2.48M | 5.49M |
| ResNet32(+6AC) | 0.47M | 1.91M | 2.48M | 8.5M |
| ResNet32(+12AC) | 0.47M | 3.14M | 2.48M | 13.62M |
| ResNet18(+6AC) | 11.23M | 24.59M | 73.34M | 128.75M |
| VGG19(+10AC) | 39.33M | 41.63M | 184.71M | 194.23M |
| VGG19(+18AC) | 39.33M | 45.42M | 184.71M | 210.03M |
| ViT-base(+11AC) | 85.88M | 86.72M | 353.14M | 362.63M |

Table 7: Training times (hours) for different AC setups from Table 3 on CIFAR100.

| | CIFAR100x5 | | | | | | | | | | | CIFAR100x10 | | | | | | | | | |
|---|---|---|---|---|---|---|---|---|---|---|---|---|---|---|---|---|---|---|---|---|---|
| ACs | ANCL | BiC | ER | EWC | FT | FTEx | GD | LODE | LwF | SSIL | Avg | ANCL | BiC | ER | EWC | FT | FTEx | GD | LODE | LwF | SSIL | Avg |
| 0 | 2.1 | 1.4 | 2.4 | 1.2 | 1.1 | 1.3 | 0.6 | 3.0 | 1.2 | 1.7 | 1.6 | 2.8 | 2.0 | 2.5 | 1.4 | 1.4 | 1.7 | 1.2 | 3.3 | 1.4 | 1.8 | 1.9 |
| 3 | 2.7 | 1.8 | 2.8 | 1.5 | 1.3 | 1.5 | 0.8 | 3.6 | 1.5 | 2.1 | 2.0 | 3.6 | 2.5 | 3.2 | 1.9 | 1.6 | 2.0 | 1.5 | 4.2 | 1.9 | 2.3 | 2.5 |
| 6 | 3.3 | 2.1 | 3.3 | 1.8 | 1.6 | 1.7 | 1.1 | 4.3 | 1.8 | 2.5 | 2.4 | 4.4 | 3.0 | 3.8 | 2.2 | 1.9 | 2.4 | 1.7 | 5.2 | 2.4 | 2.9 | 3.0 |
| 12 | 4.5 | 2.9 | 4.0 | 2.5 | 2.0 | 2.3 | 1.5 | 5.5 | 2.5 | 3.2 | 3.1 | 6.8 | 4.2 | 4.8 | 3.0 | 2.4 | 3.3 | 2.3 | 6.9 | 3.5 | 3.9 | 4.1 |

In our most common CIFAR100 setup with ResNet32, training overhead from introducing ACs is on average around 50%, which - while not negligible - is also not a big concern given modern hardware. **Please also keep in mind that our training code was not optimized and best-case training times for AC-based networks would be lower without the thorough evaluation and logging we used over the training phase.**

# 6 CONCLUSIONS

We have explored the potential of leveraging intermediate representations in neural networks to improve the performance and efficiency of continual learning through the use of auxiliary classifiers (ACs). Through our analysis, we confirmed that early network layers are more stable during continual learning, particularly in retaining information from older tasks. Building on this observation, we introduce ACs, lightweight classifiers trained on intermediate layers, as a novel enhancement to standard continual learning methods. Our results show that ACs not only help mitigate catastrophic forgetting by maintaining strong performance on older tasks but also foster diversity in classification, as different ACs specialize in classifying distinct data subsets. We demonstrate that integrating ACs into several established continual learning methods consistently yields superior performance compared to single-classifier models on benchmarks such as CIFAR100 and ImageNet100 across diverse model architectures such as ResNets, VGG19 and ViT-base. Additionally, the addition of the ACs enables computational savings and adaptability through dynamic inference, allowing models to maintain the accuracy of the baseline while reducing computational costs during inference by up to 70%. Our findings suggest that ACs can serve as a powerful tool in continual learning, not only enhancing performance but also offering efficient alternatives to standard methods in resource-constrained environments, where balancing accuracy and efficiency is critical.

**Reproducibility and ethics statement.** All our experiments were done on publicly available datasets with FACIL framework for easy reproducibility. We publish the anonymized version of our code at `https://anonymous.4open.science/r/cl-auxiliary-classifiers` and we will make it public upon the acceptance of the paper. Our research primarily focuses on fundamental machine learning problems and we do not identify any specific ethical concerns associated with our work; nonetheless, given the potential ramifications of machine learning technologies, we advise approaching their development and implementation with caution.

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
