# OpenReview forum: "Auxiliary Classifiers Improve Stability and Efficiency in Continual Learning"
_ICLR.cc/2025/Conference — Submitted to ICLR 2025_

### Official Review · Reviewer_JvvK · 2024-10-21

**Soundness:** 2
**Presentation:** 3
**Contribution:** 3
**Rating:** 5
**Confidence:** 5

**Summary:**

This paper investigated the stability of intermediate neural network layers during continual learning, where early network layers tend to be more stable. The authors then proposed to integrate auxiliary classifiers (ACs) into intermediate layers and ensemble them for improving continual learning. The authors then provided extensive experiments to demonstrate the effectiveness of the proposed ACs.

**Strengths:**

1. This paper is essentially well-organized and easy to follow.

2. The proposed ACs seem to be easy to implement and provide significant improvements over a range of continual learning baselines.

3. The proposed ACs may also reduce the computation through dynamic inference.

**Weaknesses:**

1. The authors claimed that “no work has yet explored the use of intermediate classifiers in the continual learning setting”. However, there are at least two papers focusing on using multiple ACs in continual learning. [1] proposed to use multiple side classifiers on the top of regularization-based methods. [2] added multiple ACs to the intermediate outputs and integrated their outputs for online continual learning.

2. The entire work is essentially based on the observations that the intermediate outputs behave differently and may outperform the final outputs in some cases. Is it possible to provide some mechanistic explanation for this phenomenon? Also, the advantages of intermediate outputs in unique accuracy (Figure 3) seem to be marginal for continual learning baselines. I'm not sure this is the main reason for the improved performance of the ACs.

3. The authors claimed that the dynamic inference can reduce the computation. Does this mean training costs and/or testing costs? From my understanding, the proposed ACs still need to train the entire model while skip some layers for inference.

4. The experiments are mainly performed with ResNet-based architectures. Do the proposed ACs also apply to the intermediate outputs of transformer-based architectures?

[1] More classifiers, less forgetting: A generic multi-classifier paradigm for incremental learning. ECCV 2020.

[2] Orchestrate Latent Expertise: Advancing Online Continual Learning with Multi-Level Supervision and Reverse Self-Distillation. CVPR 2024.

**Questions:**

Please refer to the Weaknesses.

--------------------------

I agree that this paper is essentially of borderline quality in terms of novelty and empirical contribution.

**Novelty**: This paper is based on the empirical observations that the earlier layers tend to be more stable in continual learning, which is not surprising because the earlier layers often capture more general features that are potentially shared by all tasks. Inspired by this, the authors then employ auxiliary classifiers (ACs) to improve offline class-incremental learning. As acknowledged by the authors, the idea of ACs is borrowed from the early-exit model, a widely used strategy in improving performance and efficiency of deep neural networks. Some advantageous properties of this work, such as reducing inference time, stem from the early-exit model rather than the authors’ own contribution. Also, I have provided two papers in continual learning that implement similar ACs in a parallel or sequential manner, respectively. The authors’ clarification of their differences, such as different continual learning setting, cannot convince me that the use of ACs in this work is completely novel.

**Empirical contribution**: I acknowledge that the authors have provided a lot of experiments, but increasing the amount of experimental results does not necessarily add to the quality of this work. This work is essentially based on the empirical connections between layer-wise stability and continual learning performance. It does not have theoretical basis to ensure applicability, which is a limitation, although I think it’s not a big problem for such a borderline paper. However, all analyses are limited to offline class-incremental learning and training from scratch (i.e., one of the most basic continual learning setting). All continual learning methods achieve very limited performance (e.g., less than 40% on the simple CIFAR100) and the benefits of ACs are remarkably more significant on the simplest FT, which further add the concerns of the applicability of this intuitive idea in realistic continual learning scenarios. Although the authors provide a “pre-train” state of ResNet and also provide ViT results with training from scratch, the overall performance is even worse. Further considering the limited continual learning scenarios and the extra parameter costs of the ACs, I think such empirical contribution is not significant enough.

In summary, I understand that the authors did a lot of experiments. However, “doing more experiments” does not necessarily mean that the quality is improved. These efforts only help to improve the understanding of the work. I think this work has been clearly demonstrated, but it’s novelty and empirical contribution remain (slightly lower than) a borderline quality.

---

> ### Author Response · Authors · 2024-11-19
> **Response to the Reviewer JvvK**
>
> We thank the reviewer for the time spent reviewing our paper. Below, we respond to the individual issues raised by the reviewer.
>
> > The authors claimed that “no work has yet explored the use of intermediate classifiers in the continual learning setting”. However, there are at least two papers focusing on using multiple ACs in continual learning. [1] proposed to use multiple side classifiers on the top of regularization-based methods. [2] added multiple ACs to the intermediate outputs and integrated their outputs for online continual learning.
>
> While the works mentioned by the reviewer are indeed reminiscent of our work, we would like to point out the critical differences between our paper and works [1] and [2]. [1] uses multiple classifiers on top of the feature extractor (backbone network) in an ensemble manner, while our paper attaches the classifiers to the intermediate network layers; both approaches are orthogonal and in principle could even be combined. [2] explores the setting of online continual learning, which is different from offline continual learning explored in our paper. Nonetheless, we thank the reviewer for pointing out those works. We have included them in the related works section in the updated paper alongside a clearer description of our contribution.
>
> > The entire work is essentially based on the observations that the intermediate outputs behave differently and may outperform the final outputs in some cases. Is it possible to provide some mechanistic explanation for this phenomenon? Also, the advantages of intermediate outputs in unique accuracy (Figure 3) seem to be marginal for continual learning baselines. I'm not sure this is the main reason for the improved performance of the ACs.
>
> Regarding Figure 3, we respectfully disagree that the advantages are marginal; adding the unique accuracy of all intermediate classifiers, you achieve around 10% or 8% accuracy for LwF and BiC, while base variants of those methods achieve around 29% or 43% accuracy respectively. Therefore, the combined unique accuracy of all the added ACs makes up for around ⅓ or ⅕ of the total accuracy of the base method.
>
> As for the different behavior of the ACs, those classifiers are built on top of different representations, so they will learn to operate on different kinds of features, and some of those features might be more stable across the learning phase as shown in our analysis in Section 3.1.
>
>
> > The authors claimed that the dynamic inference can reduce the computation. Does this mean training costs and/or testing costs? From my understanding, the proposed ACs still need to train the entire model while skip some layers for inference.
>
> The Reviewer's understanding is correct. Our paper focuses on utilizing ACs to improve continual learning performance and as an additional contribution, we show how our approach can reduce the inference time through dynamic selection of the classifier.
>
> We do not focus on training efficiency, and introducing the ACs increases training time depending on the number of classifiers (we added the exact times in Appendix O). The training time for our standard setup (6ACs) is roughly 50% higher, which we do not consider meaningful for offline class-incremental learning. Please also remember that we did not focus on optimizing the training time, and the training overhead could be reduced by optimizing the training code, so those times should be treated more as an upper bound.
>
> > The experiments are mainly performed with ResNet-based architectures. Do the proposed ACs also apply to the intermediate outputs of transformer-based architectures?
>
> Our paper already includes results for ViT models in Appendix J, and ACs achieve good improvements upon the baselines with ViTs. In addition, we performed additional experiments with deep VGG19 architecture in Appendix N. We hope these experiments improve the Reviewer's confidence in the robustness of our method.
>
> **Conclusion.** We hope we responded to most of the Reviewer's issues. We are open to further discussion if the Reviewer has any other questions.

---

> > ### Comment · Reviewer_JvvK · 2024-11-25
> >
> > I thank the authors for their rebuttal. After reading the rebuttal and other reviewers' comments, I think this work have many aspects to improve. Its idea is not completely novel, and the experimental analysis should be more comprehensive. Therefore, I keep my rating.

---

> > > ### Author Response · Authors · 2024-11-25
> > >
> > > We respectfully disagree with the vague comment that our idea is “not completely novel.” We have already outlined the critical differences between our work and prior studies, and we are open to further discussion if specific concerns are raised.
> > >
> > > Similarly, regarding the experimental analysis, we included all requested experiments, including the ViT analysis, which was present in the original submission.
> > >
> > > We kindly request more explicit and constructive points if the Reviewer is to dismiss our work.

---

### Official Review · Reviewer_ZoN8 · 2024-11-02

**Soundness:** 3
**Presentation:** 3
**Contribution:** 2
**Rating:** 5
**Confidence:** 3

**Summary:**

This paper investigates the stability of intermediate neural network layers and addresses the catastrophic forgetting problem in continual learning (CL) by utilizing features from these layers to train auxiliary classifiers (ACs). The proposed approach is novel and aims to enhance the robustness of existing CL methods.

**Strengths:**

**Originality:** The focus on leveraging intermediate layer features to train ACs as a means to combat catastrophic forgetting is an innovative contribution to the field.
**Quality:** The experimental results demonstrate that the proposed ACs significantly improves the performance of current CL methods, validating the effectiveness of the approach.
**Clarity:** The paper is well-organized and easy to follow.

**Weaknesses:**

1. The paper lacks a detailed analysis of time complexity and computational overhead. Specifically, how much additional time and memory are required for training and inference with the introduced ACs? This is a significant concern, as the practicality of the proposed method may be limited by increased resource requirements.
2. The description of how to train the ACs is unclear. Are the same strategies used for training all classifiers? What is the architecture of each classifier?
3. The choice of static inference, where the classifier with the maximum probability is selected, lacks further analysis and justification. More explanation is needed on this decision-making process.
4. In Figure 5, what does the x-axis labeled "cost" represent? Additionally, what value of $\lambda$ was used in the reported results for dynamic inference?

**Questions:**

1. What is the distribution of the final selected classifiers during inference?
2. The paper observes only six intermediate layers; it would be interesting to know if similar results apply to other layers as well.

---

> ### Author Response · Authors · 2024-11-19
> **Response to the Reviewer ZoN8**
>
> We are grateful for the time spent on our paper by the Reviewer. We address the Reviewer’s questions below.
>
> > The paper lacks a detailed analysis of time complexity and computational overhead. Specifically, how much additional time and memory are required for training and inference with the introduced ACs? This is a significant concern, as the practicality of the proposed method may be limited by increased resource requirements.
>
> We added the training times for AC-based networks on CIFAR100 in comparison with standard networks in Appendix O and the parameter and inference memory overhead of our method in Appendix P.
>
> While training computational complexity can be important (e.g. online continual learning) and ACs introduce training overhead, we do not see it as an important issue in offline class-incremental learning which is the focus of our work.
>
> > The description of how to train the ACs is unclear. Are the same strategies used for training all classifiers? What is the architecture of each classifier?
>
> For the detailed experimental setup, we refer the Reviewer to Appendix K, which we mention in Line #363 in the main paper. In this appendix, we describe AC architectures and the training procedure. If the Reviewer has any further more specific questions, we are open to answering them.
>
> > The choice of static inference, where the classifier with the maximum probability is selected, lacks further analysis and justification. More explanation is needed on this decision-making process.
> We already provide such analysis in Appendix D. While our method of choosing the most confident prediction is simple, we find that it performs well in practice.
>
> > In Figure 5, what does the x-axis labeled "cost" represent? Additionally, what value of λ was used in the reported results for dynamic inference?
>
> The cost reported on those plots is measured as the average FLOPs of the dynamic inference relative to the FLOPs for the base network with the same method. This is explained in L#404-407. We updated the Figure 5 caption to make it clearer.
>
> As for the question regarding $\lambda$ values used for dynamic inference plots, we evaluate lambda every 0.01. We also updated the main paper with this information (line #405).
>
> > What is the distribution of the final selected classifiers during inference?
>
> As per Reviewer's request, we updated the paper to include the distributions of selected classifiers, alongside their accuracy, in Appendix Q.
>
> > The paper observes only six intermediate layers; it would be interesting to know if similar results apply to other layers as well.
>
> We provide such analysis for 3 and 12 layers in ResNet32 in Table 3 in the main paper. We also include results with 12 layer ViT-base in Appendix J. In addition, we updated the paper with results for deeper VGG models with 10 and 18 ACs in Appendix N, where our technique also significantly outperforms the baselines. We hope those results add more confidence to the applicability of our method and overall performance evaluation with multiple scenarios and network architectures.
>
> **Conclusion.** We hope we addressed most of the Reviewers concerns, and we are open to further discussion if the Reviewer has any other questions.

---

> > ### Comment · Reviewer_ZoN8 · 2024-11-25
> >
> > Thank you for your response. I appreciate the additional clarifications and details provided in the rebuttal. However, I have the following suggestions and concerns:
> >
> > 1. Regarding  time complexity and computational overhead:
> > I respectfully disagree with the comment that time cost is not an important issue in offline class-incremental learning. For any method, it is essential to balance time overhead against the improvement in performance. I appreciate the statistics provided in Appendix O and P regarding training costs, but I strongly recommend integrating these insights into Table 3 in the main text. This would provide a more intuitive understanding of the trade-offs and benefits of the proposed approach.
> >
> > 2. On training the ACs and classifier selection:
> > Regarding how the ACs are trained and why the classifier is selected using the maximum probability, I suggest reorganizing the content between the main text and the appendix. As it stands, these choices are quite confusing when reading the main text alone. Improved structuring and clearer explanations would significantly enhance the accessibility of the method.
> >
> > Overall, while the authors have provided extensive experimental results, I feel the analysis could be more thorough. The method's design appears to be primarily driven by empirical results, and I recommend achieving a better balance between time costs and performance gains. For these reasons, I maintain my original score.

---

> > > ### Author Response · Authors · 2024-11-25
> > >
> > > 1. Regarding the training cost of the continual learning approaches, well-established methods such as BiC, ANCL or even standard experience replay significantly increase training time by either introducing additional phases of training or just using more data to train (e.g. in many implementations of experience replay, after the first task we train on batches that contain half new and half old data, oversampling from memory, which effectively doubles the training time). Our method allows one to adjust the training overhead by using fewer classifiers, and our results show that it performs robustly with only a few classifiers. **We achieve uniform improvements using 3 ACs, which adds around 25% training time overhead. This is still significantly less than the overhead caused by standard experience replay that in principle doubles the training time.** Our method allows the user to adjust and save the computation during inference will all tested CL approaches, which can provide way more savings in the long run.
> > > 2. We think we fairly presented the training times and the relation between the computational cost and accuracy of our method in the experimental results. What would the Reviewer consider a ‘good’ balance between them? This recommendation is not clear to us.
> > > 3. Could the Reviewer be more precise than saying that the analysis is not enough thorough? What is missing?
> > > 4. Regarding the overall organization of the text, we are grateful for the suggestions and we will incorporate them; however, we are only allowed one revision of the paper during the rebuttal stage and opted out to not modify the original paper structure during the rebuttal phase to simplify the discussion.

---

### Official Review · Reviewer_CJ85 · 2024-11-03

**Soundness:** 2
**Presentation:** 3
**Contribution:** 2
**Rating:** 5
**Confidence:** 4

**Summary:**

This paper aims to target catastrophic forgetting in continual learning as the problem statement. They
Introduce auxiliary classifiers (ACs) as a mechanism to improve performance in continual learning. The study provides analysis using linear probes and then proposes adding classifiers to intermediate layers, leveraging the fact that earlier layers of neural networks exhibit more stability. The results are shown with different Methods, naive fine-tuning , replay-based and regularizer based CL methods.

**Strengths:**

- Catastrophic forgetting is a key challenge in continual learning and this paper aims to address this critical issue
- The use of linear probing to assess accuracy at different network layers is interesting and offers insights
- The paper is well-organized and generally easy to follow

**Weaknesses:**

- The paper’s objective is bit ambiguous. It’s unclear whether the goal is to fully mitigate catastrophic forgetting or simply to offer additional accuracy sources through auxiliary classifiers. Because, forgetting still occurs, with the method seemingly redistributing accuracy rather than eliminating forgetting. This distinction needs clarification, particularly around Line 190, where the claim that the method is "less prone to forgetting" may need more evidence.

- Previous studies have already shown that early layers capture more generic features, while later layers capture task-specific semantics, so just early layers alone are often insufficient for reliable predictions. Further, though the paper incorporates auxiliary classifiers across layers, this approach introduces computational overhead. The lack of consistent patterns in the ablation studies also leaves it unclear how to optimally position these classifiers for a more efficient solution.

- The motivation to introduce auxiliary classifiers (ACs) stems from empirical analysis, but the results show inconsistent patterns across different continual learning methods. For instance, in replay-based methods, weights remain relatively stable even without ACs, suggesting that the benefits of ACs may not be as universal as claimed. This raises the question of whether adding classifiers could be unnecessary overhead for certain methods.

- LP works on frozen networks, however the hypothesis in Line 253, aims to train all classifiers, and the criteria changes. Training multiple classifiers concurrently may impact the final classifier's performance by diluting its specificity and potentially reducing network plasticity. Hence the training and the final classifier accuracy and the patterns learnt to make the prediction, can get affected ?

- Empirical analysis could be more detailed. There’s limited discussion on the scalability of this method to larger networks or more extended task sequences. The claim of reduced forgetting (Line 190) would benefit from testing on longer task sequences (>10)  and more complex (deeper) architectures. Also does the phase of training play a part, during initial epochs vs near the end of the final epochs for a task?

- Other accuracy criteria such as stability and plasticity or forward/backward transfer is not provided which are important for assessing the method's full impact on continual learning.

- Will this work when classes overlap, say in domain incremental learning?

**Questions:**

- Figure 1 is not clear, the colors blend together. In general few figures need improvement.
- Can you explain LP analysis? The classifiers at each layer are trained after the whole network is trained on all tasks and frozen?
- Line 187, is this claim correct? There are no analysis for longer tasks (more than 10)
- Can we visualize a pattern of which classifiers are being used? With multiple ACs, how is the final classifier’s predictive power affected?  Could this architecture reduce overall network plasticity?
- Line 471 -  The lack of a clear impact from varying AC numbers and positioning is surprising. This makes it difficult to form a clear intuition about the impact. Thoughts on this ablation?
- While replay and regularization methods are considered in results, parameter isolation methods such as PNN.. are not considered. Also, such as DER ++ (logit replay) are not considered?
- Line 283 - was any other criterion tried before choosing maximum confidence?
- How is threshold calculated for dynamic inference? Does it depend on arch or complexity of data or tasks?

---

> ### Author Response · Authors · 2024-11-19
> **Response to the Reviewer CJ85 (part 1)**
>
> We thank the Reviewer for assessing our work. We address the Reviewer’s questions below.
>
> > The paper’s objective is bit ambiguous. It’s unclear whether the goal is to fully mitigate catastrophic forgetting or simply to offer additional accuracy sources through auxiliary classifiers. Because, forgetting still occurs, with the method seemingly redistributing accuracy rather than eliminating forgetting. This distinction needs clarification, particularly around Line 190, where the claim that the method is "less prone to forgetting" may need more evidence.
>
> Our objective is to increase continual learning performance. We do not claim that our method is less prone to forgetting but mention that “The higher stability of early representations indicates the potential for their use in continual learning, as we can expect them to be less prone to forgetting.”, which is supported by our analysis in Section 3.1.
>
> > Previous studies have already shown that early layers capture more generic features, while later layers capture task-specific semantics, so just early layers alone are often insufficient for reliable predictions. Further, though the paper incorporates auxiliary classifiers across layers, this approach introduces computational overhead. The lack of consistent patterns in the ablation studies also leaves it unclear how to optimally position these classifiers for a more efficient solution.
>
> We do not claim that the higher stability of early representations is our main finding. Our main aim is to leverage this stability to improve continual learning performance. Our analysis in Section 3.2 shows that intermediate features can indeed be used to learn classifiers that in some cases even outperform the classifier learned with standard approach, especially on previous tasks classes.
> While our approach introduces an overhead, we also show that it consistently improves the continual learning performance.
>
> Optimal placement of ACs is a very complex problem that we consider beyond the scope of our work, and tuning this placement would require orders of magnitude more computation than we have available. We never claim to come up with any “optimal” solution to this problem, and even though we acknowledge our results are likely lower than in the ideal case, our approach still achieves consistent improvements in all the tested scenarios.
>
> > The motivation to introduce auxiliary classifiers (ACs) stems from empirical analysis, but the results show inconsistent patterns across different continual learning methods. For instance, in replay-based methods, weights remain relatively stable even without ACs, suggesting that the  benefits of ACs may not be as universal as claimed. This raises the question of whether adding classifiers could be unnecessary overhead for certain methods.
>
> We provide a comparison with non-AC methods for all our settings and show robust improvements from our methods. In our experiments, replay-based methods such as ER or LODE show the best improvements (see Table 1).
> We are interested in the Reviewer’s comment that weights remain “relatively stable” even without ACs in the replay-based method. Can we ask the Reviewier to provide some references about this?
>
> > LP works on frozen networks, however the hypothesis in Line 253, aims to train all classifiers, and the criteria changes. Training multiple classifiers concurrently may impact the final classifier's performance by diluting its specificity and potentially reducing network plasticity. Hence the training and the final classifier accuracy and the patterns learnt to make the prediction, can get affected?
>
> We provide this kind of analysis in Figure 4, where we compare the accuracy of intermediate ACs when trained (with gradient propagation) with the linear probing (which produces the final classifier identical to the no-AC network). As per this analysis, adding the ACs and their training does not hurt the final classifier performance.
>
> > Empirical analysis could be more detailed. There’s limited discussion on the scalability of this method to larger networks or more extended task sequences. The claim of reduced forgetting (Line 190) would benefit from testing on longer task sequences (>10) and more complex (deeper) architectures.
>
> We updated the paper with results on 20 and 50 task sequences for CIFAR100(see Appendix L). Additionally, we updated the paper with new results for the deeper CCN network VGG19 (see Appendix N).
>
> We would also like to point out that our paper already includes results with ViT-base on ImageNet in Appendix J (which is also mentioned in lines #461-462 of the main paper). We hope those results satisfy the Reviewer and add more confidence to the results of our method and overall performance evaluation with multiple scenarios and network architectures.

---

> ### Author Response · Authors · 2024-11-19
> **Response to the Reviewer CJ85 (part 2)**
>
> > Also does the phase of training play a part, during initial epochs vs near the end of the final epochs for a task?
>
> We do focus on evaluating the final models obtained with each method after the training finishes. The model will have different performance at different stages of the training, but this is not something unique to our method and we consider the training dynamics beyond the scope of our work that focuses on offline class-incremental learning. In online class-incremental learning, this can be of more importance.
>
> > Other accuracy criteria such as stability and plasticity or forward/backward transfer is not provided which are important for assessing the method's full impact on continual learning.
>
> Forgetting or stability in the context of our method are hard to analyze, as we use multiple classifiers that can override the decision of each other. Our analysis in Section 3 considers the stability of network representations in continual learning and shows that ACs maintain better performance on older data than the final classifier and to a degree learn to specialize on small subsets of data. The combination of all those factors leads to better performance of AC networks in continual learning, as they are able to provide some degree of redundancy on older data which helps alleviate the forgetting as compared with the single-classifier case.
>
> > Will this work when classes overlap, say in domain incremental learning?
>
> Yes, in principle our approach should work for domain-incremental learning, but its performance will be heavily dependent on the performance of the base method (e.g. LwF). We believe that the robustness introduced by ACs should translate to gains in this setting as well, but we consider such evaluation beyond the scope of our work.
>
> > Figure 1 is not clear, the colors blend together. In general few figures need improvement.
>
> We updated the colormap in Figure 2 and similar Figures. As for the heatmap plots (e.g. Figure 1) we do not see any problems with their design, nor did the other two reviewers who praised our paper’s readability. We are open to further changing other Figures, should the Reviewer provide us with more detailed instructions on what could be improved.
>
> > Can you explain LP analysis? The classifiers at each layer are trained after the whole network is trained on all tasks and frozen?
>
> We explain the linear probing setup in lines #196-200 and the experimental setup in detail in Appendix K. Yes, linear probing classifiers are trained after each task on top of a frozen trained network.
>
> > Line 187, is this claim correct? There are no analysis for longer tasks (more than 10)
>
> The claim that we observe more stability is supported by CKA analysis in Section 3 and better overall results with our method across all the settings we evaluated.
>
> Following the Reviewer's advice, we extended our experiments to longer task sequences (20 and 50 tasks on CIFAR100) in Appendix L and added the evaluation for VGG19 network in Appendix N. Our method also performs well in such circumstances.
>
> > Can we visualize a pattern of which classifiers are being used?
>
> We updated the paper with classifier selection patterns and their accuracy in Appendix Q.
>
> > With multiple ACs, how is the final classifier’s predictive power affected? Could this architecture reduce overall network plasticity?
>
> We provide this kind of analysis in Figure 4, where we compare the accuracy of intermediate ACs with gradient propagation with linear probing. In this setting, the final classifier is identical to the classifier in a standard network without ACs. As per this analysis, the addition of the ACs and their training does not hurt the final classifier performance.
>
> As for plasticity, we believe our method should increase overall plasticity, as ACs can learn different patterns in intermediate network representations.
>
> > Line 471 - The lack of a clear impact from varying AC numbers and positioning is surprising. This makes it difficult to form a clear intuition about the impact. Thoughts on this ablation?
>
> We do not see this as much of an issue, but rather as proof of the robustness of our idea. Inspired by the Reviewer’s question, we also include leave-one-out ablation on the placement of the AC in Appendix R . We do not claim that our AC placement is optimal, which do not hide and even explicitly state in L#375-377. While our results are not conclusive on the placement of the AC, we see consistent improvements with our method. Our results can be considered as a lower bound for future works that would optimize AC placement for best continual learning performance, but the complexity of such a problem makes it beyond the scope of our work.

---

> ### Author Response · Authors · 2024-11-19
> **Response to the Reviewer CJ85 (part 3)**
>
> > While replay and regularization methods are considered in results, parameter isolation methods such as PNN.. are not considered. Also, such as DER ++ (logit replay) are not considered?
>
> Parameter isolation methods are usually more suitable to task-incremental learning and require architectural changes, so we opted to not use them in our analysis due to our paper focus being on class-incremental learning and our codebase being built on top of FACIL framework that does not focus on such methods. In principle, task-agnostic architectural methods or methods such as DER++ could be used with our approach.
>
> Following the Reviewer's advice, we implemented DER++ in FACIL and evaluated it with ACs in Appendix M. As in the case of all other tested methods, the addition of ACs improves the performance of DER++.
>
> > Line 283 - was any other criterion tried before choosing maximum confidence?
>
> Yes, we have an analysis on this in Appendix D (as already pointed out in line #310).
>
> > How is threshold calculated for dynamic inference? Does it depend on arch or complexity of data or tasks?
>
> The threshold is not calculated in any way, it is a hyperparameter for dynamic inference and the user has to tune the threshold to match his desired objective (e.g. whether the goal is to keep the original accuracy while reducing compute or maintain a given performance threshold). As evidenced in our analysis of dynamic inference for different settings and networks, the cost-accuracy characteristics are not universal and depend on both data and model architecture.
>
> **Conclusion.** We hope we addressed most of the Reviewers' concerns regarding the robustness of our idea and are open to further discussion with the Reviewer.

---

> ### Author Response · Authors · 2024-11-27
>
> We appreciate that the Reviewer increased their score (3 -> 5) for our work after our responses. However, as we believe our detailed rebuttal has addressed most, if not all, of the concerns raised in the initial review, we would greatly value further clarification on why the Reviewer still considers the work below the acceptance threshold. We feel that the effort we have put into providing a thorough response deserves a more detailed and precise explanation.

---

### Author Response · Authors · 2024-11-19
**Summary response**

We thank all the Reviwers for the time spent on our work and their comments that helped us improve the paper.

We updated the paper according to the Reviewers' suggestions (changes marked with olive color) and provided additional results such as experiments with longer (20 and 50) task sequences, deeper architectures (VGG19), and additional baseline in DER++. In addition, wherever suitable, we tried to point the Reviewers to sections in the Appendix of the original version of the paper that were initially overlooked by them (e.g. experimental details and results for Vision Transformers). We hope our answers can improve the Reviewers’ opinion about the robustness and quality of our work.

We uploaded an updated paper and appendix to OpenReview. For ease of discussion, we add all the new results to the end of the appendices during the discussion phase, but we are open to reorganizing the sections in the paper later following the Reviewers' advice.

---

### Author Response · Authors · 2024-11-27
**Summary of discussion and changes to the paper during the rebuttal so far**

We have finalized the revisions to our work during the rebuttal phase, which improved the quality of our submission. We have provided thorough responses to each Reviewer and addressed the raised concerns. However, despite our efforts, we are disappointed by the Reviewers' unwillingness to engage in the discussion and re-evaluate the scores for our submission after improvements, especially given its initial borderline rating.

Below, we summarize all the content added during the rebuttal for the reviewers and readers:
* We revised the structure of the paper to improve its clarity and flow.
* We included additional results demonstrating the robustness of our method on longer task sequences.
* We added evaluation for a deeper convolutional network VGG19.
* We incorporated an additional baseline comparison with DER++.
* We moved the results with ViT models requested by the Reviewers from the appendix to the main paper. We highlight that these results were already present in the appendix of the original submission.
* We provided further details on the training and inference processes for the models, addressing Reviewers' queries.
* We added classifier selection statistics from our experiments and provided additional ablation on the classifier placement.
* We slightly updated the related works section to include the works highlighted by the Reviewers.

In a separate comment below, we also address the common concerns expressed by the Reviewers. We kindly ask the Reviewers to consider reevaluating our paperwork and engage in a more thoughtful discussion, as we feel our effort invested in both the paper and the rebuttal warrants a fair and respectful response.

---

> ### Author Response · Authors · 2024-11-27
> **Joint response to common concerns**
>
> Below, we collectively address the common concerns mentioned by the Reviewers, as we believe these were not fairly evaluated or adequately considered.
>
> **Simplicity of our method.** While the idea behind our approach is straightforward, it is well-supported by our representational stability analysis and empirical results across various settings, models, and continual learning methods where we consistently outperform the standard methods. We view the simplicity of our method as a strength that enables its easy implementation and adoption.
>
> **Primarily empirical motivation.** Our method is motivated by an extensive analysis of intermediate layer representations. We find critiques about the empirical motivation behind our work vague, especially in the context of a machine learning conference, where much of the research is inherently empirically driven.
>
> **Lack of theory behind AC placement**. Determining the placement of intermediate classifiers is a highly complex issue, as demonstrated by numerous studies in the early-exit field (we refer the Reviewers to an early-exit method survey [1] that demonstrates the sheer amount of work dedicated to this problem). Since our paper focuses on continual learning, we deliberately opted for a simple approach to AC architecture and placement to not obfuscate our evaluation. **Despite the simplicity of our approach, our method already achieves consistent improvements across all settings without any extensive AC placement optimization, further underscoring its robustness.**
>
> **Overhead of our method**. While we acknowledge that incorporating ACs introduces additional overhead, in turn, **ACs offer significant computational savings during inference**, which we view as more important in offline continual learning settings. During inference, AC-enhanced models can achieve performance comparable to the original method while using only 30-40% of the compute. The training time and memory overhead introduced by our method are also relatively modest and depend on the model architecture. Larger models like ViTs exhibit significantly lower overhead compared to smaller models, and, realistically, the overhead is more important in the case of those larger models. In our experiments with ResNet32 (**which is worst-case scenario in our evaluation**), the standard setup with 6 ACs results in a 50% increase in training time, and this overhead could be further decreased through code optimizations and more efficient AC placement. In comparison, well-established replay-based continual learning methods, such as experience replay, incur higher training overhead just due to repeatedly processing more data during training. Therefore, **we do not consider a training overhead of our method to be a realistic drawback in modern continual learning scenarios.** The memory overhead caused by the addition of ACs is likewise a concern mostly for smaller CNNs and is negligible in the case of larger models like ViT-base.
>
> **Insufficient analysis behind our method.** Our work includes extensive ablation studies and comprehensive experiments, including new ones requested by the Reviewers. If there are any additional concerns, we are more than willing to address them; however, we kindly request more specific feedback to do so effectively.
>
> **Insufficient experiments.** We evaluate our idea across 8 settings (CIFAR100 split into 5/10/20/50 even tasks, ImageNet100 split into 5/10 even tasks, CIFAR100 50 task warm-start with 5/10 tasks), 11 methods (FT, FT+Ex, GDumb, ANCL, BiC, ER, DER++, EWC, LODE, LwF, SSIL) and 3 different model architectures (ResNet32/18, VGG19, ViT-base). Our evaluation is more extensive than most continual learning papers, and **our method robustly performs across all the tested settings.**
>
> **Insufficient method ablations.** We provide ablation studies about the classifier placement, architecture, number of classifiers, and exit rule. In addition, we extensively compare linear probing with classifier training. Again, we kindly ask for more precise feedback from the Reviewers.
>
> References:
> [1] Rahmath P, Haseena, et al. "Early-Exit Deep Neural Network-A Comprehensive Survey." ACM Computing Surveys (2022).

---

### Author Response · Authors · 2024-12-04

We are deeply disappointed with the quality of the reviews and the discussion phase at ICLR 2025. The reviews we have received were very low-effort and overlooked a lot of our work, and despite all the efforts we made during the rebuttal the reviewers refused to engage with us in any meaningful discussion. This was especially disheartening as our work was initially rated borderline, with reviewers posing numerous questions we thoroughly addressed. Although the conference extended the discussion period by a week, this additional time led to no engagement from the reviewers, leaving us frustrated and disheartened. Such disregard for any meaningful dialogue feels deeply disrespectful to the effort we invested in responding to the reviewers — if discussion is not taken seriously by the reviewers, why pose all those questions in the first place? Our experience at ICLR 2025 raises serious concerns about the fairness and rigor of the review process, which falls far below the standards expected of a such conference.

---

### Meta-Review · Area_Chair_Tp9m · 2024-12-22

**Metareview:**

The paper proposes to utilize Auxiliary Classifiers (AC) to benefit continual learning, particularly offline class-incremental learning. The proposed method is simple -- attach AC to the intermediate layers so that more stable generic features could contribute to improved continual learning performance. The empirical results show the benefits of the proposed method. However, despite the positive empirical results, they were mostly done on small scale datasets, CIFAR100 and ImageNet100, which make it less certain about the scalability of the proposed method. Particularly, in modern ML applications with much larger computation and data scale, it would be **necessary to show the benefit of the proposed method on larger scale dataset**, at least at the level of ImageNet100, for offline class incremental learning setting. (Several previous work, e.g., SSIL, already has results on those setting.) Moreover, the gain the proposed method achieves tends to diminish as the base method accuracy increases, so again it is not clear about how much benefit the proposed method will bring provided the increased computational costs. To that end, AC believes the current submission is not sufficient for a publication at ICLR, yet, and the decision is Reject.

**Additional Comments On Reviewer Discussion:**

The reviewers were actively engaged in the rebuttal process. Most of the reviewers mentioned that the method -- attaching ACs to the intermediate layers -- is relatively straightforward and the empirical results are not very convincing as the readers cannot clearly draw when the AC would be truly beneficial across diverse architecture and datasets.

**JvvK** mentioned about the novelty and less rigorous aspects of the empirical results.
**CJ85** asked about how to proceed with AC losses when the base CL algorithms have multiple loss functions, which should be an important consideration in practice, and the authors have not responded.

---

### Decision · Program_Chairs · 2025-01-22

Reject